
# The unusually long cold spell and the snowstorm Filomena in Spain in January 2021

Philipp Zschenderlein[1] and Heini Wernli[1]

[1]Institute for Atmospheric and Climate Science, ETH Zurich, Universitätstr. 16, 8092 Zurich, Switzerland

**Correspondence:** philipp.zschenderlein@env.ethz.ch

**Abstract.** In early January 2021, Spain was affected by two extreme events – an unusually long cold spell and a heavy snowfall event associated with extratropical cyclone Filomena. For example, up to 50 cm of snow fell in Madrid and the surrounding areas in 4 days. Already during 9 days prior to the snowfall event, anomalously cold temperatures at 850 hPa and night frosts prevailed over large parts of Spain. During this period, anomalously cold and dry air was transported towards Spain from central Europe and even from the Barents Sea. The storm Filomena, which was responsible for major parts of the snowfall event, developed from a precursor low-pressure system over the central North Atlantic. Filomena intensified due to interaction with an upper-level potential vorticity (PV) trough, which was the result of anticyclonic wave breaking over Europe. In turn, this wave breaking was related to an intense surface anticyclone and upper-level ridge, whose formation was strongly influenced by a warm conveyor belt outflow of a cyclone off the coast of Newfoundland. The most intense snowfall occurred on 09 January and was associated with a sharp air mass boundary with an equivalent potential temperature difference at 850 hPa across Spain exceeding 20 K. Overall, the combination of pre-existing cold surface temperatures, the optimal position of the air mass boundary, and the dynamical forcing for ascent induced by Filomena and its associated upper-level trough were all essential – and in parts physically independent – ingredients for this extreme snowfall event to occur.

## 1 Introduction

Extreme weather events in winter like cold spells and heavy snowfalls often have hazardous impacts on society and economy. Such an event occurred in Spain in early January 2021, when large parts of the country were affected by a long-lasting cold spell and an unusually strong snowfall event associated with extratropical cyclone Filomena. The cold spell commenced at the end of December 2020 and from 07 to 10 January 2021, large parts of Spain were covered by a thick snow layer. The heavy snowfall has led to four deaths and to more than 1.8 billion EUR of economic costs (AON, 2021). According to a report issued by AEMET (2021), the intense snowfall and widespread snow cover in Spain associated with Filomena were exceptional. Many mountain areas registered deep accumulations of snow, but also areas typically untroubled by strong snowfall, like Madrid, were heavily affected. Barajas International Airport measured 38 cm of snow and parts of Madrid registered more than 50 cm





of snow, while most of the precipitation in the Canary Islands and southern Spain fell as rain. Directly after the snowstorm,
daily minimum temperatures reached record low values. On 12 January 2021, minimum temperatures down to -26°C were
measured in Torremocha de Jiloca – a municipality located between Madrid and Valencia (AEMET, 2021). Snowstorms occur
rather rarely in the Mediterranean, it is therefore worth analysing the processes that led to the long-lasting cold spell and the
heavy snowfall associated with Filomena.

The synoptic dynamics of heavy snowfalls have been frequently studied for events in the US, in particular for events along
the east coast. A famous example of such an event is the Presidents' Day snowstorm in February 1979 that affected large parts
of the US east coast. It was the result of a rapidly deepening extratropical cyclone that propagated along an intense coastal
front (Bosart, 1981). At upper levels, the interaction of subtropical and polar jet streaks further enhanced the development of
the surface cyclone (Uccellini et al., 1984). However, also climatologically warmer regions, like the southeastern US, can be
affected by snowstorms. In that region, snowstorms are accompanied by extratropical cyclones that are often located at the
right entrance of the jet (Mote et al., 1997) – an area of the polar jet streak that is typically associated with strong forcing for
ascent [e.g., Fig. 3 in Uccellini and Kocin (1987)], which favours precipitation and cloud formation. In addition to the listed
factors, advection of moisture and low static stability appear to be also relevant (O'Hara et al., 2009). When lower-tropospheric
temperatures are slightly above the freezing point, the falling snow starts to melt and is therefore very wet and heavy. Such wet
snowfall events occurred, for example, in November 2005 over parts of Germany (Frick and Wernli, 2012) and in March 2010
in northeastern Spain (Llasat et al., 2014).

In winter, the Mediterranean receives its largest fraction of annual precipitation when it is frequently influenced by extratrop-
ical cyclones (Trigo et al., 1999; Wernli and Schwierz, 2006). In this region and season, cyclones and fronts are often involved
in extreme precipitation events (Field and Wood, 2007; Pfahl and Wernli, 2012; Catto and Pfahl, 2013), however, they rarely
lead to extreme snowfalls due to climatologically warm temperatures in the Mediterranean (Tayanç et al., 1998; Llasat et al.,
2014; Gascón et al., 2015; de Pablo Dávila et al., 2021). In the region of Madrid, precipitation typically occurs as rain also in
January, due to surface temperatures above the freezing level (DWD, 2021). Hence, an extreme snowfall event and a persistent
snow layer at the surface need at least two essential ingredients: (i) cold surface temperatures, and (ii) a (slow-moving) cyclone
to produce a substantial amount of accumulated precipitation.

The first essential ingredient, cold surface temperatures, prevents the snow from melting when reaching the surface. Cold
spells in the Iberian Peninsula are typically caused by advection of cold continental air from the northeast, which is favoured
by an anticyclone over the British Isles and a cut-off low over Italy (Santos et al., 2015). Cold air is typically quite dry and
therefore advection and lifting of slightly warmer and moister air, ideally above the still very cold near-surface layer, is needed
to produce large amounts of snow. Heavy snowfall events in the Iberian Peninsula are associated with an upper-level trough
over southwestern Europe and a surface cyclone located either over the Mediterranean Sea or central Europe (Esteban et al.,
2005; de Pablo Dávila et al., 2021). This large-scale pattern facilitates the advection of anomalously cold air from the north
near the surface and of warm and humid air from the south towards the snowfall area above the cold layer (Llasat et al., 2014;
Gascón et al., 2015).





Warm and humid air in cyclones is typically associated with warm conveyor belts (WCBs), which are coherent, poleward ascending airstreams in extratropical cyclones (e.g., Browning et al., 1973; Wernli and Davies, 1997). The ascent from the
planetary boundary layer towards the upper troposphere leads to a cooling of the air parcels and the condensed water vapour leads to the formation of clouds and precipitation (Browning, 1990; Joos and Wernli, 2012). WCBs frequently contribute to heavy precipitation events (Grams et al., 2014; Pfahl et al., 2014) and can therefore also contribute to heavy snowfall events (Gehring et al., 2020). In addition to precipitation, WCBs also influence the synoptic-scale flow in the upper troposphere, e.g., the amplification of Rossby waves (Massacand et al., 2001; Grams et al., 2011; Raveh-Rubin and Flaounas, 2017), and the
formation of blocks (Pfahl et al., 2015).

In this study, we focus on the synoptic-scale dynamic evolution of the cold spell and the snowstorm Filomena in early January 2021. We will not analyse mesoscale processes, which can locally intensify snowfall, e.g., via the formation of mesoscale snow bands (e.g., Nicosia and Grumm, 1999; Schumacher et al., 2010). The key questions of our case study analysis are:

1. How unusual was the cold spell and which processes led to the anomalously low temperatures over Spain?

2. Which processes led to the formation of Filomena?

3. Which characteristics of Filomena facilitated the heavy snowfall?

Furthermore, we compare this event to earlier heavy snowfall events in the Iberian Peninsula.

Section 2 provides an overview of the employed data and methods. Then, Sect. 3 presents the evolution of temperature and precipitation in the first half of January 2021. Processes leading to the cold spell are discussed in Sect. 4 with the help
of trajectories (question 1). Subsequently, Sect. 5 investigates the evolution of cyclone Filomena and the large-scale weather situation in the North Atlantic (question 2), as well as the heavy snowfall event (question 3). In Sect. 6, we compare the snowstorm Filomena with two other snowstorms in the Iberian Peninsula. Finally, Sect. 7 concludes and discusses our main findings.

## 2 Data and Methods

### 2.1 Data


This study is based on hourly ERA5 reanalyses (Hersbach et al., 2020) from the European Centre for Medium-Range Weather Forecasts (ECMWF) interpolated to a $0.5° \times 0.5°$ longitude-latitude grid for the period 1979-2021. Secondary fields like equivalent potential temperature and potential vorticity (PV) were calculated from the model level data. Total precipitation, i.e. the sum of large-scale and convective precipitation, as well as snowfall data are taken from short-term ECMWF forecasts.

### 2.2 Trajectories


To investigate the origin of the cold spell, ten-day backward trajectories, driven by three-dimensional wind fields on vertical model levels, are calculated with LAGRANTO (Wernli and Davies, 1997; Sprenger and Wernli, 2015). Trajectories are





initialised only from grid points in the Iberian Peninsula that were affected by the snowstorm, i.e. grid points where the accumulated snowfall between 00 UTC 07 January and 00 UTC 11 January 2021 exceeded 5 mm (Fig. 1a). From these grid points, trajectories are started every 3 hours during the cold-spell period from 00 UTC 30 December 2020 to 00 UTC 09 January 2021 at 10, 30, 50, 70, and 90 hPa above ground level, similar to, e.g., Zschenderlein et al. (2019) and Papritz (2020). Physical parameters traced along the trajectories include temperature, potential temperature, and specific humidity. Note that the cold spell lasted beyond 9 January, but since we are primarily interested in the origin of the cold air prior to the peak of the heavy snowfall, we have only calculated trajectories until this date. This trajectory analysis for the event in January 2021 is complemented by a climatological analysis, in which trajectories are initialised at the same grid points and vertical levels, and on the same December and January days in all previous winters from 1980 to 2020.

Trajectories are also used to identify WCBs over the North Atlantic and the Iberian Peninsula during the study period in early January 2021 and for other heavy snowfall events. For that, we computed 2-day forward trajectories initialised between 20°-50°N and 90°W-15°E in the lower troposphere from 1000 to 700 hPa in 25 hPa increments. Trajectories ascending more than 600 hPa in 2 days were selected as WCBs, and in addition, we identify the ascent phase of a WCB, which corresponds to WCB trajectories in the pressure layer between 800 and 400 hPa, following the definition of Madonna et al. (2014).

## 2.3 Identification of further heavy snowfall events and cold spells

In order to compare the snowstorm Filomena with other snowstorms in the Iberian Peninsula, we identify events in all winters from 1979 to 2019. For that, we compute the accumulated snowfall in the Iberian Peninsula at grid points over land between 36°N-44°N and 10°W-4°E in a 4-day window and determine the area affected by snowfall larger than 5 mm. We use a time window of 4 days because the snowfall event in early January 2021 lasted for 4 days. In addition, we identify cold spells in all winters from 1979 to 2019, but only for the area where the accumulated snowfall between 00 UTC 07 January and 00 UTC 11 January 2021 exceeded 5 mm (Fig. 1a). In this area, we average the daily 2-m minimum temperature and define a cold spell as a continuous period with daily 2-m minimum temperatures below 0 °C.

## 3 Precipitation and temperature evolution in Spain in early January 2021

Many areas in Spain and Morocco experienced large amounts of precipitation between 07 and 10 January 2021. Most of it fell as rain in southern Spain and Morocco, and as snow in the rest of Spain and in the Atlas Mountains (Fig. 1a). The highest snow accumulation occurred near Madrid, as well as in eastern and southeastern Spain. Precipitation was confined to a corridor extending from Morocco, Andalusia, Madrid to provinces along the east coast of Spain, while Portugal, France and other parts of North Africa were not affected by the event. Overall, precipitation was most intense in Andalusia and off the Moroccan coast (more than 75 mm in 4 days), where it however fell exclusively as rain. Compared to all winters from 1979-2019 in the Iberian Peninsula, the 4-day snow period associated with Filomena has, according to ERA5, the highest recorded snow accumulation and is one of the events with the largest extent (Fig. 1b).





The area enclosed by the 5 mm snowfall contour in Fig. 1a (thereafter called snowfall area) experienced night frosts since
30 December 2020 and negative 2-m and 850 hPa temperature anomalies (Fig. 2a). In addition, the period was mostly dry
and only two short precipitation events on 28 December 2020 and 01 January 2021 affected the region prior to the snowstorm
(Fig. 2b). The precipitation between 07 and 10 January was associated with cyclone Filomena and, in contrast to the two earlier
precipitation events, most of the precipitation fell as snow. The diurnal cycle of temperatures on 09 January, i.e. at the peak of
the snowfall event, was less pronounced than during the other days. After the snow event, night-time temperatures dropped, but
at 850 hPa, a new air mass arrived and temperatures at this level were increasing again (Fig. 2a). Due to the existing snow cover,
2-m temperatures during night were still anomalously low and an inversion formed. Interestingly, the 2-m temperature and its
anomaly curve have an opposite diurnal cycle, which means that the low day temperatures were particularly anomalous. As an
example on 09 January, daily minimum temperature was about 3 K colder than climatology, while daily maximum temperature
was more than 7 K colder than climatology, which is presumably related to the lack of incoming solar radiation. The cold spell,
i.e. the period with daily minimum temperatures below the freezing point, lasted until 19 January 2021 (not shown).

Due to the cold spell in early January, the surface progressively cooled down and this prevented the snow from melting
when reaching the surface on 07-10 January. Therefore, this cold spell preconditioning appears to be an important ingredient
of the extreme snowfall accumulation. In the next section, we analyse the origin of this cold spell and put its duration in a
climatological context.

**4   Processes leading to the long-lasting cold spell**

Before discussing the origin of the cold air near the surface over Spain in early January 2021, we first quantify how anomalous
the duration of the cold spell was. We therefore compare the duration of this event with previous cold spells in the winters
1979 to 2019 that occurred in the area affected by snowfall on 07-10 January 2021 (Fig. 1a). Figure 3 shows the number of
continuous days with minimum temperatures below 0 °C at 2 m and 850 hPa, respectively. The cold spell in early January
140   2021 lasted for 21 days at 2 m, and for 14 days at 850 hPa. The difference is presumably associated with the inversion that
formed after the snow event due to warm air advection at 850 hPa (Fig. 2a). The longevity of the cold spell at the surface can
be classified as extreme because it exceeds the 99th percentile of the climatology of continuous night frost days at 2 m, while
the duration of the cold spell at 850 hPa is slightly below the 99th percentile (Fig. 3). The 99th percentile of the climatological
cold spell duration at 2 m is 16 days, and with 21 days the January 2021 event is the 3rd longest in the period 1979-2019. The
145   longest cold spell with 29 days occurred from 29 December 1999 to 26 January 2000 and the second longest from 10 January
1992 to 04 February 1992. Both events were not followed or accompanied by heavy snowfalls, hence, the combination of
long-lasting cold temperatures and high accumulations of snow is a unique feature of the event in early January 2021, at least
since 1979.

Figure 4 illustrates the origin of the cold air in the snowfall area during the cold spell period from 30 December 2020 to 09
January 2021, based on the trajectory method described in Section 2.2. The locations of air parcels are shown as frequencies
per km$^6$, i.e. the trajectory count at each grid point is normalised by the number of all trajectories, such that the spatial integral



yields 100 %. The normalisation was applied separately to the trajectories of the cold spell and the climatology. The purple lines in Fig. 4 show the position of air parcels three days prior to their arrival over Spain in early January 2021. Most of them are located over central Europe, France, the British Isles, and some even over the Barents Sea (Fig. 4). In comparison with

climatology, i.e. with the origin of air on all days between 30 December and 09 January in the years 1980 to 2020, the origin of the cold air in January 2021 is unusually far north. Transport of cold air from the north has been reported as an important feature of cold spells in general (Bieli et al., 2015).

   The air mass origin in 2021 is not only unusual when considering the position three days prior to arrival in Spain, but also for the whole 10-day period of the backward trajectories (Fig. 5). Cold air parcels in 2021 are advected from further

north (Fig. 5a), and from further east (Fig. 5b); they originate from slightly lower altitudes (Fig. 5c) and are transported along substantially lower isentropes compared to climatology (Fig. 5e). The lower isentrope (median value of 285 K compared to 295 K in the climatology) is related to the northern origin. This anomalously cold air is also drier than normal (Fig. 5f). It is interesting that the temperature of the air parcels at their origin is only slightly colder than normal (Fig. 5d), which can be explained by compensating effects of being further north (Fig. 5a) and at a lower altitude (Fig. 5c). During the last five days

of the transport, they were less warmed compared to climatology and therefore stayed below 0 °C (Fig. 5d). Physically, air parcels change their temperatures due to adiabatic processes (vertical motion), or diabatic processes, e.g. radiative cooling or heating by surface sensible heat fluxes. Potential temperature changes along the backward trajectories associated with the cold spell are weak and temperature changes are therefore primarily due to adiabatic warming during descent. Indeed, air parcels leading to the cold spell descend less than climatologically expected (Fig. 5c). Therefore, the cold air is eventually cold due to

(i) slightly colder origin temperatures, and (ii) weaker subsidence and therefore less adiabatic warming. Hence, over a period of approximately ten days, cold and dry air was steadily advected towards the snowfall area and helped to progressively cool down the surface. This trajectory analysis explains why 2-m temperatures in Spain were exceptionally cold in early January 2021. We now continue with investigating the lifecycle of cyclone Filomena, which was responsible for the snowfall event at the end of this exceptional cold spell.

**5  Snowstorm Filomena**

Cyclone Filomena evolved from a decaying low-pressure system over the central to eastern North Atlantic (marked with L1 in Fig. 6c). It is therefore interesting to first discuss the evolution of this precursor cyclone L1. The cyclogenesis of L1 occurred at 12 UTC 02 January near Newfoundland (Fig. 6a). The weak upper-level PV trough intensified into a narrow PV streamer until 00 UTC 04 January 2021, and L1 underwent rapid intensification reaching a core pressure of nearly 980 hPa in the western

North Atlantic (Fig. 6b). Within the next hours, the elongated PV streamer broke up into a PV cut-off. Until 00 UTC 06 January, L1 weakened and propagated further east, but remained at fairly low latitudes (36°N, Fig. 6c). The large-scale flow pattern over the North Atlantic then quickly changed in the next 36 hours. A second low pressure system, L2, evolved near Newfoundland at almost the same location as L1 three days earlier (Fig. 6a,c). This low was associated with strong warm conveyor belt activity at its leading edge (Fig. 6d). This warm conveyor belt helped to intensify the downstream ridge and the





associated surface high-pressure system over the central North Atlantic, a situation that is often observed in this region (Grams et al., 2011). The strong ridge contributes to anticyclonic wave breaking over western Europe, which is important for the re-intensification of the decaying low L1 and the cyclogenesis of Filomena near 32°N at 12 UTC 07 January 2021 (marked with F in Fig. 6d). The anticyclonic wave breaking pushed the upper-level PV trough towards the Iberian Peninsula, and therefore, Filomena slightly intensified while propagating north-eastwards towards the Iberian Peninsula (Fig. 6e). At 00 UTC 09 January

2021, Filomena was located over the Iberian Peninsula where it led to intense rain and snow in this area (Fig. 6f, cf. Fig. 1a and 2b). In addition to the anticyclonic wave breaking, the transverse vertical circulations associated with the subtropical and the polar jet influenced the development of Filomena and the associated warm conveyor belt. For example at 00 UTC 09 January 2021, Filomena is located at the intersection of the left exit region of the subtropical jet and the right entrance of the polar jet stream (Fig. S1). A similar situation was observed for snowstorms in the eastern US (Uccellini and Kocin, 1987) and for

cyclogenesis in the eastern Mediterranean (Prezerakos et al., 2006).

The storm Filomena led to strong contrasts of equivalent potential temperature ($\theta_e$) at 850 hPa exceeding 20 K over Spain at the peak of the snowfall event at 00 UTC 09 January 2021 (Fig. 7c). Four days earlier, on 05 January 2021, the air was cold and dry over the Iberian Peninsula with $\theta_e$ values below 300 K (Fig. 7a). With the approach of Filomena from 07-09 January 2021, $\theta_e$ values in northern Spain decreased even more and over Northern Africa, $\theta_e$ values increased due to the approaching

warm sector of Filomena with values exceeding 310 K (Fig. 7b,c). Most of the precipitation occurred in areas with very large $\theta_e$ gradients, i.e. fronts. The warm front on 09 January led to the strongest snowfall in central Spain associated with a warm conveyor belt, which ascended exactly at the location of strong snowfalls (Fig. 6d-f). The extreme accumulation of snow is also related to the slow propagation of Filomena and its warm front ahead of the quasi-stationary upper-level trough. Figure 8 illustrates the stationarity of the front with the aid of meridionally oriented vertical cross sections of $\theta_e$ and cloud variables. At

all three timesteps, the upper part of the clouds are completely glaciated, at medium levels liquid and ice clouds coexist, while the lower part consists of cloud water with rain on the warm side of the front and (intense) snow on the cold side. On the warm side of the front clouds were quite shallow, while on the cold side they were comparably deep with cloud tops up to 350 hPa. South of the front, the air in the warm conveyor belt inflow was very moist. It is nicely visible that the ascent of the warm conveyor belt at the warm front is co-located with the heaviest snowfall at all three time steps. Altogether, the combination of

a slow-moving front, sufficient lower-tropospheric moisture on its warm side, and pre-existing cold surface temperatures prior to and during the snowstorm were essential ingredients for the large amounts of surface snow accumulation.

## 6  Comparison with other heavy snowfall events

In the final part of this study, snowstorm Filomena is compared to the second and third strongest snowstorms in the period 1979-2019, based on the method described in Section 2.3. These events are labelled as E2 and E3 in Fig. 1b, respectively, and

we briefly compare the snow cover, the synoptic situation, and the air mass origin with the Filomena event in 2021. For this section, all Figures can be found in the supplement.




The second strongest snowstorm, the E2 event, occurred from 02 to 05 January 1997 and mainly affected mountainous regions in Northern Spain, in particular the Pyrenees (Fig. S2a). Hence, the snowstorm was associated with significant orographic effects, in contrast to snowstorm Filomena. Another difference to Filomena was the higher surface temperature, which led to
stronger snowmelt in the lowlands. Similar to Filomena, temperatures decreased before the onset of the snowfall, however, with still higher temperatures than during Filomena (not shown). The third strongest snowstorm, the E3 event, lasted from 28 to 31 January 1986 and also affected mountainous regions (Fig. S2b). Cold air from Greenland was transported and moistened over the North Atlantic, and when reaching northern Spain, it was orographically lifted and contributed to the cloud and precipitation formation (not shown). East of the Pyrenees, an intense cyclone with a core pressure of about 970 hPa enabled
the strong snowfall (Fig. S3). In both events, the horizontal $\theta_e$ contrasts were weaker than during the passage of Filomena (Figs. S4a, S4b, and 7c). It is interesting that the very deep cyclone during the E3 event is accompanied by weaker $\theta_e$ gradients than the weak cyclone Filomena with core pressures of about 1000 hPa (Fig. 6f). High $\theta_e$ gradients can influence the intensity of a cyclone, as for example the winter storms hitting Europe in December 1999, which were very intense (Ulbrich et al., 2001). However, it appears that the strong $\theta_e$ gradient during the passage of Filomena did not have a large impact on the
intensification of the cyclone, but rather on the precipitation intensity. In comparison with events E2 and E3, it is remarkable that the accumulation of snow during Filomena is so large without substantial orographic influences from the Pyrenees, which underlines the importance of the cold spell preconditioning in early January 2021.

## 7 Conclusions

In this study, we analysed the synoptic-scale dynamic evolution of two extreme events in early January 2021 affecting large
parts of the Iberian Peninsula – a long-lasting cold spell from 30 December 2020 to 19 January 2021, and a heavy snowfall event from 07 to 10 January associated with the passage of extratropical cyclone Filomena. The snowstorm was exceptional due to the widespread snow cover and large accumulations of snow even in areas like Madrid that are rarely troubled by snow. Prior to the snowfall, a cold spell established and served as an important preconditioning for the snowfall event. We now summarise the key results by addressing the research questions raised in the Introduction.

1. How unusual was the cold spell and which processes led to the anomalously low temperatures?

The cold spell lasted 21 days and exceeded the 99th percentile of the number of continuous days with 2-m minimum temperature below 0 °C. Due to the steady advection of dry and cold air from the north, surface temperatures could progressively cool down, which later prevented the snow from melting. A comparison with the climatological origin of near-surface air over Spain revealed that during the cold spell, exceptionally cold and dry air was transported from
central and western Europe, but also from the Barents Sea. This air mass origin is shown to be unusual.

2. Which processes led to the formation of Filomena?

Filomena evolved from a decaying low in the central North Atlantic at fairly low latitudes and intensified along a quasi-stationary upper-level PV trough, which formed due to anticyclonic Rossby wave breaking over Europe. This wave



breaking in turn was related to an intense anticyclone over the central North Atlantic, which was strongly influenced by the WCB outflow of an intense extratropical cyclone off the coast of Newfoundland. Filomena was located at the intersection of the left exit of the subtropical jet and the right entrance of the polar jet – both regions are associated with large-scale forcing for ascent, and therefore, the configuration of the two jets further influenced the intensification of Filomena. However, Filomena was not very intense with a minimum pressure of about 1000 hPa. This sequence of events, including the upstream influence from a cyclone and its warm conveyor belt in the western North Atlantic illustrates the complex dynamics responsible for the formation of Filomena.

3. Which characteristics of Filomena facilitated the heavy snowfall?

Large parts of Spain, especially the area in and around Madrid, were affected by intense snowfall that occurred along a sharp air mass boundary identified by a horizontal contrast of $\theta_e$ of more than 20 K across Spain. Along the warm frontal part of this boundary, dynamical forcing for ascent, imposed by the surface cyclone and upper-level trough, led to warm conveyor belt ascent and intense snowfall. Hence, the combination of pre-existing cold near-surface temperatures, the "optimal" position of the air mass boundary, and the ascent of the moist air along steep moist isentropes associated with the warm front of Filomena were essential ingredients for this extreme snowfall event to occur.

This case study reveals how different processes on various time scales co-produced this hazardous event. In addition, this event is also an example of the relevance of diabatic processes for large-scale weather patterns in the North Atlantic (e.g. Grams et al., 2011; Magnusson, 2017). A comparison with two heavy snowfall events and two long-lasting cold spells has revealed the uniqueness of the events in early January 2021. First, snowfall events with comparable accumulated 4-day snowfalls as during Filomena in the Iberian Peninsula are usually strongly influenced by orographic effects, i.e. the Pyrenees typically record the highest snow accumulation. In such situations, northwesterly winds transport cold and humid air towards the Pyrenees and produce precipitation by orographic lifting. Second, the co-occurrence of such a long-lasting cold spell and heavy snowfall is unusual – the two longest-lasting cold spells in that region were not associated with significant amounts of snow. Overall, our case study analysis, together with the comparison with other heavy snowfall events and cold spells, show the special characteristics of this events in early January 2021.

*Data availability.* ERA5 data can be downloaded from https://cds.climate.copernicus.eu (last access: 17 December 2021).

*Author contributions.* PZ designed the study, performed all analysis, and discussed all results with HW. PZ wrote the manuscript with feedback from HW.





*Acknowledgements.* The authors acknowledge funding of the INTEXseas project from the European Research Council (ERC) under the European Union's Horizon 2020 research and innovation programme (grant agreement no. 787652).





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




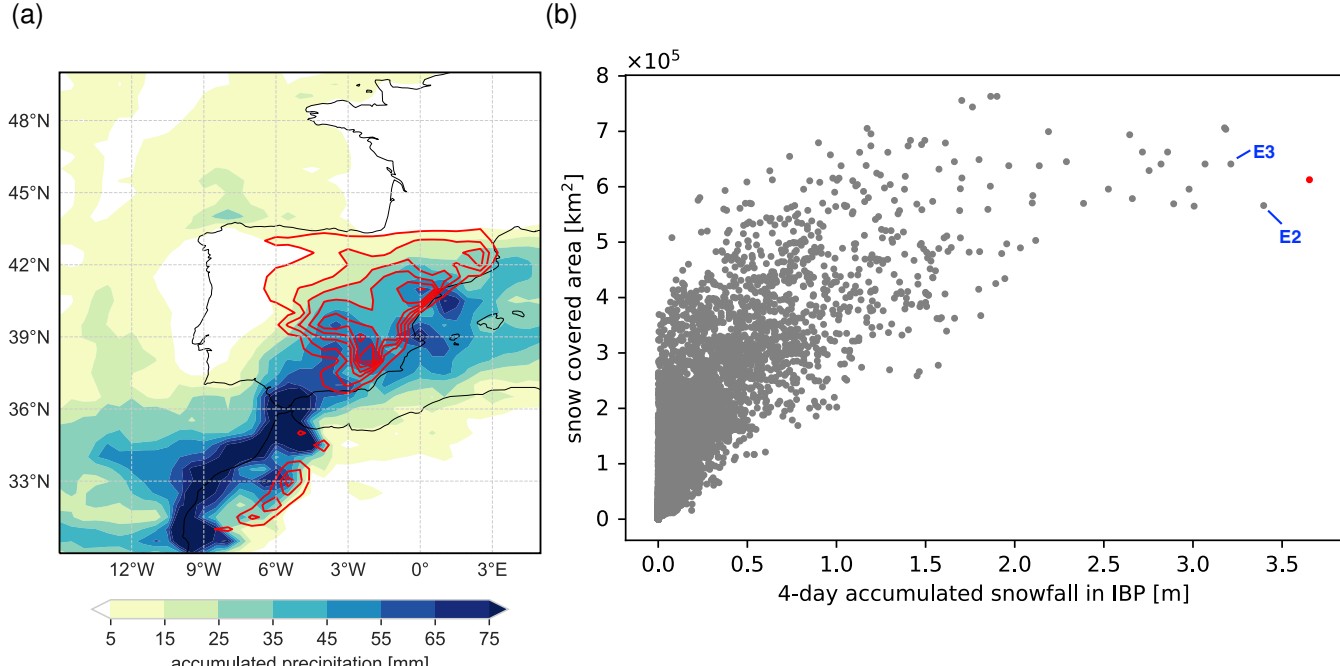

**Figure 1.** (a) Precipitation accumulated from 00 UTC 07 January to 00 UTC 11 January 2021 (from ECMWF short-range forecasts). The red contours denote the accumulated snowfall in this period, starting from 5 mm in 10 mm increments. (b) 4-day accumulated snowfall vs. snow covered area of all winters from 1979 to 2019 in the Iberian Peninsula (IBP), based on the method described in Sect. 2.3. The red point represents the period 07-10 January 2021 (Filomena), label E2 marks the period 02-05 January 1997 and E3 the period 28-31 January 1986.

**Figure 2.** (a) Temporal evolution of 2-m temperature (solid) and 850-hPa temperature (dashed) over Spain from 27 December 2020 to 13 January 2021. Grey lines show anomalies. (b) Temporal evolution of precipitation (blue) and snowfall (red). All values are averaged over the area enclosed by the 5 mm snowfall contour in Fig. 1a.




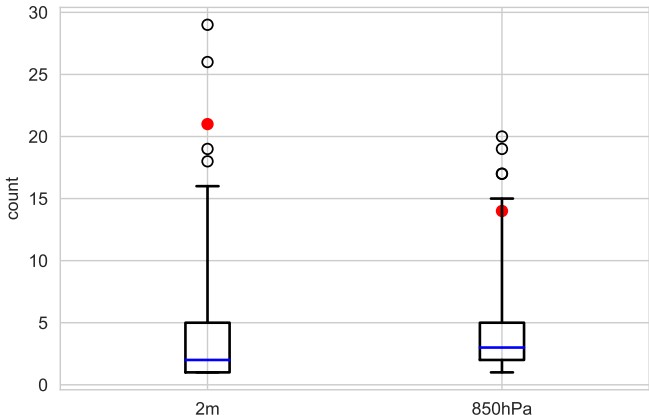

**Figure 3.** Boxplot of the duration of cold spells over Spain in the winters 1979-2019. Values are number of days with continuous minimum temperatures below 0 °C at 2 m and 850 hPa, respectively, in the area enclosed by the 5 mm snowfall contour in Fig. 1a. The blue lines denote the median, the boxes the inter-quartile range, the whiskers the 1st and 99th percentiles, and the open circles the outliers. The red circles denote the cold spell duration in early January 2021.


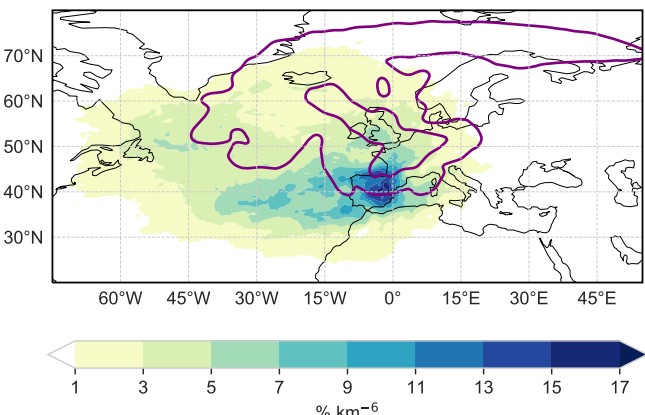

**Figure 4.** Location of air parcels three days prior to their arrival in the snowfall area over Spain (see text for details). The colour shading represents the climatological origin of near-surface air in late December and early January, and the purple lines (1 and $17\,\%\,\mathrm{km}^{-6}$) represent the origin of air associated with the cold spell in late December 2020 and early January 2021.


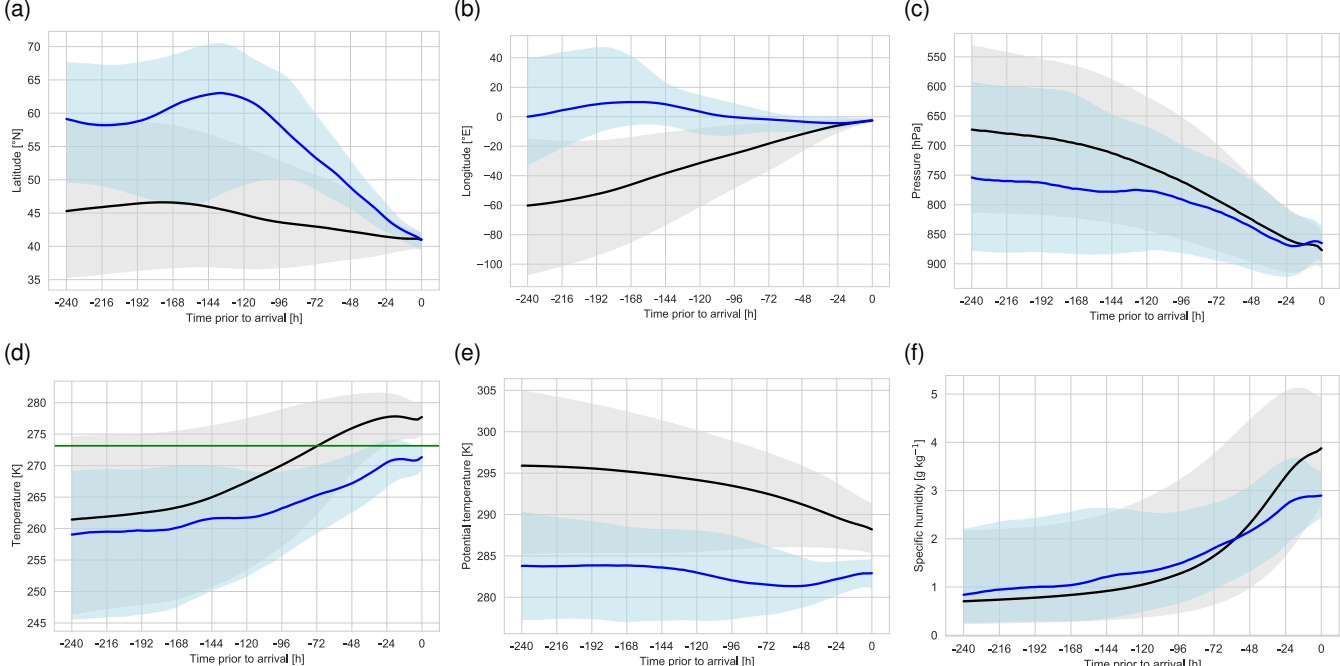

**Figure 5.** Time series along backward trajectories from Spain of (a-c) latitude, longitude, and pressure and (d-f) temperature, potential temperature and specific humidity. The grey colours denote values for the climatology and the blue colours for the cold spell in early January 2021. The bold lines denote the median and the shading the inter-quartile range. The green line in (d) marks a temperature of $0\,°C$.

**Figure 6.** Synoptic maps with sea level pressure (black), 1-h accumulated total precipitation (blue contours for 1 mm), 1-h accumulated snowfall (red contours for 1 mm), PV at 320 K in colour shading, and the position of every 30th WCB trajectory in the ascent phase. The panels are valid at (a) 12 UTC 02 January, (b) 00 UTC 04 January, (c) 00 UTC 06 January, (d) 12 UTC 07 January, (e) 00 UTC 08 January, and (f) 00 UTC 09 January 2021.




**Figure 7.** Maps with equivalent potential temperature ($\theta_e$) at 850 hPa (colour shading), 1-h total accumulated precipitation (blue contours for 1 mm), and 1-h accumulated snowfall (red contours for 1 mm). Panels are valid at (a) 00 UTC 05 January, (b) 00 UTC 07 January, (c) 00 UTC 09 January, and (d) 00 UTC 10 January 2021. The grey lines denote the anomaly of $\theta_e$ at 850 hPa (solid: +4K, dashed: -4K), which is calculated with respect to a 21-day running mean.


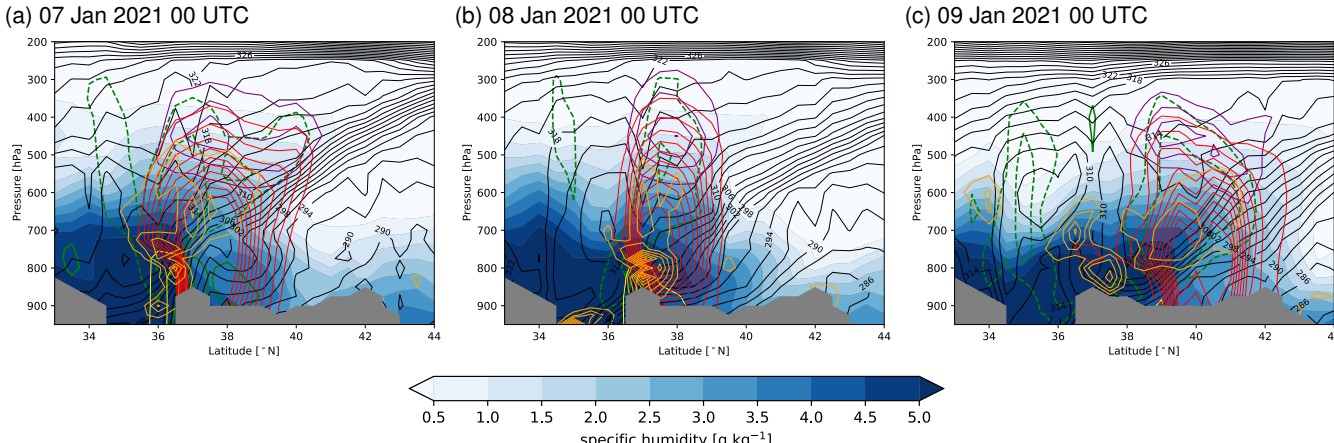

**Figure 8.** Vertical cross sections from 33 °N to 44 °N along 3 °W at (a) 00 UTC 07 January, (b) 00 UTC 08 January, and (c) 00 UTC 09 January 2021. Shown are equivalent potential temperature (black), specific humidity (blue shading), vertical motion (green; for -1, -0.5, 0.5, and 1 Pa s$^{-1}$), rain water content (yellow), ice water content (purple), snow water content (red), and liquid water content (orange). The latter four variables are shown in intervals between 0.05 g kg$^{-1}$ and 0.55 g kg$^{-1}$ in 0.05 g kg$^{-1}$ increments.