# Peer review of "The unusually long cold spell and the snowstorm Filomena in Spain in January 2021"

_Natural Hazards and Earth System Sciences, 2021_

## Referee Comment (RC1)

The unusually long cold spell and the snowstorm Filomena in Spain in January 2021
NHESS-2021-396

This paper aims to determine the processes that led to a long cold spell and heavy snowfall event in Spain in January 2021.  The authors pose 3 key questions; (i) how unusual were these events; (ii) which processes led to the formation of extratropical cyclone Filomena; and (iiii) which characteristics of Filomena facilitated the heavy snowfall.  By comparison with climatology, they provide evidence to show that the events were rare.  However, with respects to the development mechanisms of Filomena there are some conclusions that require more supporting evidence (see general comments 3 & 4 and specific comments 9 & 10).

General comments
1.   Lines 131, 142,172 and 242.  In several places it is stated that the 'surface progressively cooled down'.  However, I couldn't find any surface temperature observations in the paper to support this statement.  Are the authors referring to the 2m temperatures?  If so, these are not 'surface' temperatures. For clarity I also suggest using 'air frost' to avoid ambiguity with ground frosts.
2.   Lines 147, 266 and 271.  On several occasions reference is made to the 'uniqueness', 'unique feature' or 'special characteristics' of the event.  However, I do not think that analysis of 2 events with different features confirms uniqueness.  Have the authors looked for combination of 'long-lasting cold temperatures and high accumulations of snow' over their 40-year dataset to see if they occur during other periods?
3.   Line 184 and elsewhere.  I understand that anticyclonic outflow from warm conveyor belts have been shown to increase the amplitude of downstream ridges in previous studies, but I couldn't find any evidence in this paper to suggest this is what happened in this case study.  Since this statement is repeated in the abstract and conclusions, I think there should be strong evidence to support this statement for the case study analysed.
4.   Lines 186 and elsewhere.  It is stated that the strong ridge contributes to anticyclonic wave breaking.  Perhaps I missed this in the paper, but how are the authors defining anticyclonic wave breaking?  Also, it is concluded that wave breaking is important for the re-intensification of the decaying low L1 and the cyclogenesis of Filomena. This is repeated in the abstract and conclusions, but it wasn't clear how the authors reached this conclusion.

Specific comments
1.   Abstract. The authors include an example of 'up to 50 cm of snow fell in Madrid and the surrounding area in 4-days'.  This has little context as motivation.  In section 3 they state that this was the highest recorded 4-day snow accumulation event in the last 40 years. Perhaps this could be added to the abstract to provide context?
2.   Line 25.  What records were broken during this event?  How long is the daily minimum temperature record?
3.   Line 27. The motivation for this work could be broader.  Currently, the motivation give is that 'it is worth analysing the processes that led to the long-lasting cold spell and heavy snowfall' because it is 'rather rare in the Mediterranean'.  While I agree that it is scientifically interesting to analyse rare events when they occur, I wonder if there may be other reasons for performing the analysis.  For example, the authors do not comment on the skill of the NWP forecast for this event. Was it poorly forecast or well forecast?  What can we learn from a single extreme event?
4.   Lines 46-47.  In the text 'at least two essential ingredients' are given for large snow accumulations.  I always understood that temperatures < 2°C throughout the depth of

the atmosphere through which snow was falling was also necessary criteria for snow to remain frozen? Also, it seems odd that a moderate precipitation rate is not included as a criterion since precipitation accumulation is the product of precipitation rate and duration.

5. Line 105.  Is the ranking of this event sensitive to the choice of a 4-day window?
6. Line 129.  It is 'presumed' that incoming solar radiation is reduced during the period 9-10 January 2021.  This hypothesis should be relatively easy to test.
7. Lines 162 and 244.  It is difficult to conclude from figure 5f that the airmass was drier than normal since there is significant overlap of the interquartile range over the 10-day back trajectories.
8. Line 183.  How is the strength of the warm conveyor belt activity quantified?  Is it the areal extent of the grey shading in figure 6?
9. Line 229.  How do the authors know that the strong theta_e gradient did not have a large impact on the intensification of the cyclone?  I could not find the argument leading to this conclusion.
10. Line 262. I was not convinced that the stated conditions are 'essential ingredients' for this extreme snowfall event since the counterfactual experiment has not been demonstrated using either numerical modelling, or at least similar events with 'missing' ingredients analysed.

Typographical errors
1. Line 208. Could 'nicely visible' simply be 'visible'?
2. Line 27.  I think 'rather rarely' could simply be 'rarely'.

---

## Referee Comment (RC2)

[referee-annotated manuscript omitted]

---

## Author Comment (AC1)

NHESS-2021-396

**The unusually long cold spell and the snowstorm Filomena in Spain in January 2021**

Response to the Reviewers' comments by Philipp Zschenderlein and Heini Wernli

We thank the three reviewers for their detailed and critical feedback. While some reviewers appreciated our investigation of upstream processes leading to the snowstorm (Rossby wave breaking, warm conveyor belts), the overall feedback of the reviewers is fairly negative. In particular reviewer 3 concludes that "the study does not contribute any new knowledge" and reviewer 2 pointed us to a detailed report about the event issued by the Agencia Estatal de Meteorología (AEMET), in Spanish language, which indeed highlights already many of our key findings (importance of cold spell preconditioning, unusual track of cyclone Filomena). Although we don't think that there is enough dynamical insight about Mediterranean snowstorms published in the English peer-reviewed literature, we accept the critique of lack of novelty and therefore will not submit a revised manuscript to NHESS.

Nevertheless, in these final author comments, we reply to all points raised by the reviewers and we indicate how one potentially could have further improved the manuscript. Our replies are written in blue and the comments of the reviewers in black colour.

**Reviewer 1**

This paper aims to determine the processes that led to a long cold spell and heavy snowfall event in Spain in January 2021. The authors pose 3 key questions; (i) how unusual were these events; (ii) which processes led to the formation of extratropical cyclone Filomena; and (iiii) which characteristics of Filomena facilitated the heavy snowfall. By comparison with climatology, they provide evidence to show that the events were rare. However, with respects to the development mechanisms of Filomena there are some conclusions that require more supporting evidence (see general comments 3 & 4 and specific comments 9 & 10).

General comments

1. Lines 131, 142,172 and 242. In several places it is stated that the 'surface progressively cooled down'. However, I couldn't find any surface temperature observations in the paper to support this statement. Are the authors referring to the 2m temperatures? If so, these are not 'surface' temperatures. For clarity I also suggest using 'air frost' to avoid ambiguity with ground frosts.

We will change the wording to "near-surface temperature" and "air frost". Yes, we are using 2-m temperatures from ERA5 throughout the whole study.

2. Lines 147, 266 and 271. On several occasions reference is made to the 'uniqueness', 'unique feature' or 'special characteristics' of the event. However, I do not think that analysis of 2 events with different features confirms uniqueness. Have the authors looked for combination of 'long-lasting cold temperatures and high accumulations of snow' over their 40-year dataset to see if they occur during other periods?

No, we have not specifically looked at the combination of long-lasting cold temperatures and high accumulations of snow. We identified the two longest-lasting cold spells in ERA5 (according to our definition) and found no substantial accumulations of snow. However, it would be interesting to identify events with both long-lasting cold temperatures and high accumulations of snow. You are right that we should avoid using the term "unique", which will be removed in the revised manuscript.

3. Line 184 and elsewhere. I understand that anticyclonic outflow from warm conveyor belts have been shown to increase the amplitude of downstream ridges in previous studies, but I couldn't find any evidence in this paper to suggest this is what happened in this case study. Since this statement is repeated in the abstract and conclusions, I think there should be strong evidence to support this statement for the case study analysed.

Many thanks for this comment. You are right that this point is still a bit weak in the current manuscript. We will therefore add more text and/or figures to provide more evidence, e.g., by showing the location of warm conveyor belt outflows.

4. Lines 186 and elsewhere. It is stated that the strong ridge contributes to anticyclonic wave breaking. Perhaps I missed this in the paper, but how are the authors defining anticyclonic wave breaking? Also, it is concluded that wave breaking is important for the re-intensification of the decaying low L1 and the cyclogenesis of Filomena. This is repeated in the abstract and conclusions, but it wasn't clear how the authors reached this conclusion.

We identify anticyclonic wave breaking based on PV at 320 K in Fig. 6. We will describe the aspect of the anticyclonic wave breaking more thoroughly in the revised paper.

Specific comments
1. Abstract. The authors include an example of 'up to 50 cm of snow fell in Madrid and the surrounding area in 4-days'. This has little context as motivation. In section 3 they state that this was the highest recorded 4-day snow accumulation event in the last 40 years. Perhaps this could be added to the abstract to provide context?

Many thanks for this suggestion. We included the 50 cm of snow in Madrid because it was mentioned in a technical report issued by AEMET. We will remove this example and instead mention that the snowstorm was associated with the highest recorded 4-day snow accumulation event in the last 40 years (according to ERA5).

2. Line 25. What records were broken during this event? How long is the daily minimum temperature record?

Reviewer 2 pointed us to the full report by AEMET (the AEMET report that we found when writing our paper was only a short version). In this report, they mention several record low temperatures. In a revised version, we will add more details from the full AEMET report.

3. Line 27. The motivation for this work could be broader. Currently, the motivation give is that 'it is worth analysing the processes that led to the long-lasting cold spell and heavy snowfall' because it is 'rather rare in the Mediterranean'. While I agree that it is scientifically

interesting to analyse rare events when they occur, I wonder if there may be other reasons for performing the analysis. For example, the authors do not comment on the skill of the NWP forecast for this event. Was it poorly forecast or well forecast? What can we learn from a single extreme event?

According to a report issued by AEMET, the event was quite well forecast. We will mention this in the Introduction. An interesting aspect of this case study was the coincidence of processes on different time scales, i.e., the long-lasting cold spell and the shorter lasting snowstorm. Understanding such events could be helpful in refining forecasting procedures: cold spell predictions at sub-seasonal time scale might indicate the potential for such events, but short-term predictions of the cyclone are then required to adequately forecast the event. We will add this remark to the Introduction. In addition, we will specify the argument that heavy snowfalls are "rather rare in the Mediterranean". In the revised version, we will expand the comparison with climatology and with other cases. We could therefore better assess, why areas typically untroubled by heavy snowfalls, e.g., Madrid, were associated with high accumulations of snow. As suggested by reviewer 3, adding a full analysis of other heavy snowfall events would provide more substance to the single extreme event analysis.

4. Lines 46-47. In the text 'at least two essential ingredients' are given for large snow accumulations. I always understood that temperatures < 2°C throughout the depth of the atmosphere through which snow was falling was also necessary criteria for snow to remain frozen? Also, it seems odd that a moderate precipitation rate is not included as a criterion since precipitation accumulation is the product of precipitation rate and duration.

Many thanks for these valuable suggestions, which we will include in the Introduction.

5. Line 105. Is the ranking of this event sensitive to the choice of a 4-day window?

We could check this.

6. Line 129. It is 'presumed' that incoming solar radiation is reduced during the period 9-10 January 2021. This hypothesis should be relatively easy to test.

We could check this.

7. Lines 162 and 244. It is difficult to conclude from figure 5f that the airmass was drier than normal since there is significant overlap of the interquartile range over the 10-day back trajectories.

You are right that over most parts of the 10-day period, both air masses have significant overlap in terms of specific humidity, however, during the last day, the air leading to the cold spell is drier than climatology. We will therefore be more specific in our arguments.

8. Line 183. How is the strength of the warm conveyor belt activity quantified? Is it the areal extent of the grey shading in figure 6?

We have not quantified the strength of the warm conveyor belt activity. But the areal extent of the grey crosses in Fig. 6 would be a good measure. We will therefore remove the word

"strong" and compare the areal extent of the warm conveyor belt ascent with other lows in Fig. 6.

9. Line 229. How do the authors know that the strong theta_e gradient did not have a large impact on the intensification of the cyclone? I could not find the argument leading to this conclusion.

This sentence is indeed misleading. We will therefore remove this sentence.

10. Line 262. I was not convinced that the stated conditions are 'essential ingredients' for this extreme snowfall event since the counterfactual experiment has not been demonstrated using either numerical modelling, or at least similar events with 'missing' ingredients analysed.

This is an important point that should be improved in the revised manuscript. We could analyse other heavy snowfall events and long-lasting cold spells, and probably the combination of both (as suggested by Reviewer 3), which would give clearer indications of the "essential ingredients".

Typographical errors
1. Line 208. Could 'nicely visible' simply be 'visible'?
We will change this.
2. Line 27. I think 'rather rarely' could simply be 'rarely'.
We will change this.

**Reviewer 2**

This paper analyzes the dynamical and physical aspects on a synoptic scale that contribute to the intense snowfall and cold spell associated with the extratropical cyclone Filomena, which affected the Canary Islands and the Iberian Peninsula at the beginning of January 2021.

The authors pose three questions and the answers give substance to the article. The questions are:
- How unusual was the cold wave and what processes led to the abnormally low temperatures in Spain?
- What processes led to the formation of Filomena?
- What characteristics of Filomena facilitated the heavy snowfall?

General comments:
The authors cite a complete technical report issued by the Spanish National Weather Service (AEMET) after this episode that gives answer to the above-mentioned questions:
http://www.aemet.es/es/conocermas/recursos_en_linea/publicaciones_y_estudios/estudios/detalles/informe_filomena_ola_de_frio
Nevertheless, I think that the analysis of dynamical aspects, identifying PV streamers, ridge and trough amplification, Rossby wave rupture, and WCBs is interesting and could justify the intensification of the cyclone after the tropical interaction.

We thank the reviewer for pointing us to the full report by AEMET, of which we were not aware when we submitted our paper. We were citing a shorter report by AEMET (http://www.aemet.es/en/conocermas/borrascas/2020-2021/estudios_e_impactos/filomena). The full report by AEMET gives an overview of the synoptic conditions leading to Filomena and nicely summarises the warnings issued by AEMET. In addition, it lists several observations, e.g., snow height, accumulated precipitation, and low temperatures. Clearly, this detailed report has to be prominently cited in the revised version of our manuscript.

The reviewer comments that all questions raised by us are answered in the technical report by AEMET. Although the (non-peer-reviewed) report summarises the synoptical situation and lists observed quantities, we believe that we provide a more dynamical perspective on this case study. In addition, with our approach, we can identify other cases and can compare them to the Filomena event, using the same tools. However, we thank the reviewer for mentioning this full report. We will therefore include more details from this report in the Introduction, in particular the forecast aspect, and will more clearly state our goals.

One difference to the technical report by AEMET is that we are not only analysing one single event but compare this event to other heavy snowfall events and with climatology. And for that, we need a dataset, which is spatially, temporally, and physically consistent. In our view, this is fulfilled with ERA5. The reviewer often suggests using observational datasets, e.g., station data or soundings from radiosondes. However, we do not want to include observational datasets, because:

(a) They are not temporally and spatially uniform, which is important for the comparison with other cases and/or climatology.
(b) We are not interested in mesoscale processes that are not well captured by ERA5 reanalysis.
(c) Observations are typically only valid for one location, but since we are interested in the larger synoptic-scale picture, we prefer using ERA5 reanalysis.

The ingredients necessary for heavy snowfalls in the interior of the Iberian Peninsula are well known, cold air at low levels and humid and warm air advection. The sources of cold air mass advection over Iberian Peninsula are clearly identified too, so I think that the analysis of back trajectories does not seem to be of great interest (except for the identification of WCBs). However, the present work, based in the reanalysis and climatology of ERA-5, serves to justify the rarity of this anomalous and exceptional episode.

As suggested by the two other reviewers, we could analyse other heavy snowfall events and long-lasting cold spells, and probably the combination of both events. For this, we will also use trajectory analysis. We will then decide whether to include the trajectory analysis or not. And, in general, if a review mentions that something is "well known" then it would be appropriate to provide some references. We really did not find a lot of peer-reviewed literature about the ingredients for heavy snowfall in Iberia.

It is worth mentioned that the IFS and HARMONIE models successfully predicted this historic snowfall event well in advance, in the same way, the Spanish National Weather Service (AEMET) issued the corresponding forecasts and warnings well in advance.

Thank you for this information. We will include this in the Introduction.

The mesoscale characteristics, such as the complex topography of the affected area or the thermal and humidity profile in low layers were decisive in this episode, but in this work no reference is made to the snow level (around 500 m) or to the orography.

As noted in the Introduction, we will not analyse mesoscale characteristics of this event. Our main aim is to analyse the synoptic-scale processes leading to this event and to compare this with climatology. Therefore, we need reanalysis data, in our case ERA5.

Likewise, the time interval selected for the study (between 07 and 10 January) does not seem to be the most appropriate, since there were precipitations and snowfalls in the Iberian Peninsula between 06 and 10 January, and the intense snowfalls occurred on 08 and 09 January.

We agree that also on 06 January, there was precipitation in the Iberian Peninsula. However, according to ERA5, snowfall started on 07 January.

I think that more information should be given on the climatic characteristics of the month of January in the study area. In the months of December and January the nocturnal frosts are frequent (the period between 06 and 10 January is climatologically the lowest minimum temperature period, and at Adolfo Suarez-Barajas Madrid airport, the average number of frost days in January is 8). These frosts occur not only due to cold advection of polar or arctic air masses, but also to subsequent surface radiative processes (long nights, calm or light winds and clear skies).

Thank you for this comment. We will include these aspects in the Introduction.

AEMET defines a cold spell not just an episode with minimum temperatures below 0 C in a wide area, it applies a much more demanding criteria. In this sense, according to AEMET's technical report on Filomena, only the subsequent period to Filomena is considered a significant cold spell.

Thank you for this comment, we will note that AEMET uses a stricter definition. However, we use our definition for the climatological comparison, and we will therefore not change our definition.

Numerical model reanalysis are a useful tool, but they cannot substitute for observational data. ERA-5 has a horizontal resolution of 35 km and 37 levels in the vertical, compared to 9 km and 137 levels of the IFS-HRES model or 2.5 km and 65 vertical levels respectively of the HARMONIE limited area model. I strongly recommend the use of real observational data, taking into account the dense network of surface meteorological stations and the soundings available in the study area.

Thank you for this comment, but as stated above and in the Introduction of the manuscript, we are not analysing mesoscale processes. Since we aim to compare the case study with other events and with climatology, we think that ERA5 is a very good dataset for our study.

Regarding the adverse impact of Filomena, there were barely mentioned some effects that were very significant in the Canary Islands (intense rainfall and strong gusts of wind). The heavy snowfalls occurred on 08 and 09 January, but there was notable rainfall the previous days (06 and 07 January) in the southeast of the Iberian Peninsula, affected by a warm front associated to Filomena. The exceptionality of the snowfall, according to the climatology of the model, is reflected in the Extreme Forecast Index (EFI) developed by the ECMWF. In this sense, the snowfall was exceptional, fundamentally in the center of the Iberian Peninsula.

Thank you for this comment. We will briefly mention other effects of Filomena. The exceptionality of the snowfall can be seen in both panels of Fig. 1, which we will emphasize more in the manuscript.

**The reviewer added comments directly in the manuscript, which he/she hasn't listed here. We therefore copied his/her comments from the manuscript to this document reply accordingly.**

L19: Two people death due to floods associated to torrential rain in Mijas (Málaga) https://www.efas.eu/en/news/storm-filomena-andalusia-region-southern-spain-january-2021 I think the other death people were two  homeless,  killed by extremely low temperatures.

We took the number of four deaths from the AON report. However, we think that we should not go into more details about the cause of the deaths.

L23: Adolfo Suarez-Barajas International Airport

We will adapt this.

L27: Snowstorms in the Mediterranean area, specifically in Spain, are not unusual, it depends on the altitude of the considered area. (Remember the complex orography of the Iberian Peninsula. (De Pablo et al. 2021)

We will mention that heavy snowfalls are not seldom at higher altitudes, but rare at lower altitudes, especially with dense population, as Madrid.

L46: Please, can you cite an official reference?
I suggest WMO or AEMET
http://www.aemet.es/es/serviciosclimaticos/datosclimatologicos/valoresclimatologicos
https://worldweather.wmo.int/en/home.html

In our view, the climate diagram from the DWD can be regarded as official.

L46: deep and persistent

We will adapt this.

L84: Please, could you explain which IFS model was used? I mean HRES or ENS.

For the whole study, we used ERA5 reanalysis. Note that precipitation in ERA5 always comes from short-range forecasts and not from hourly analysis fields.

L105: Please, could you specify what kind of IFS data were used to calculate this value? Why not use real data?

We used snowfall from ERA5. We are not using "real" data because we want to compare our results with climatology. And for that, we need spatially and temporally consistent data. Point observations are not appropriate for our study.

L104-105: The accuracy of the 2m daily minimum temperature from the model reanalysis should be specified to determine the real 2m daily minimum temperature. Why not use the real observed data?

We are not using observation data for the same reasons as in the comment above.

L116: According to AEMET´s data, the highest accumulated rainfall was 252 mm between 05 and 11 January in Estepona (Málaga), with torrential rain on 08 January (206 mm in 24 h)

Thank you for the information, but we will not include this in the manuscript, since it is not the scope of our study.

L122: There were precipitation in some meteorological observatories as Estepona (Malaga) on 05 and 06 January. According to AEMET´s weather analysis chart on 06 January, this shows a warm front associated to Filomena close to the southeast of the Iberian Peninsula, the beginning of the rainy and snowy period associated with Filomena in the Iberian Peninsula could be fixed on 06 January.

Thank you, but since we use ERA5 and not AEMETs weather analysis charts, we will not include these details.

L123-124: Why not show the meteogram of a meteorological observatory or several soundings, for example from Madrid? The daily thermal oscillation of the previous days is probably due to the radiation balance corresponding to a high plateau, with long nights, calm or light winds and clear skies (without ruling out the cold advection that occurred at the beginning of January).

As stated above, we use ERA5, and will therefore not show any observation or sounding data. This would lead to another type of study.

L128: I think that model climatology is very different than real data climatology, this should be contrasted with real observational data.

In our view, a reanalysis data is an observational dataset. ERA5 reanalyses include an enormously large set of observational data to provide the most consistent estimate of the state of the atmosphere. It is not appropriate to refer to ERA5 as a "model climatology".

L132-133: Of course, but due to the cooling effect associated with the evaporation of precipitation, with a suitable vertical profile of temperature and humidity, it is possible for snow to reach the surface with initial ground temperatures above freezing, especially with dry conditions in the air layer immediately above the surface.

Thank you for this comment, but we are not analysing vertical profiles. We just wanted to stress that the cold spell preconditioning is important for the high accumulations of snow.

L137-138: AEMET has analysed the cold spell episodes that have occurred in Spain since 1975. The criteria to define a cold spell is different, but in 2021, they identified two different periods, from 05 to 08 January and from 11 to 18 January.

http://www.aemet.es/documentos/es/conocermas/recursos_en_linea/publicaciones_y_est udios/estudios/Olas_frio/Olas_Frio_ActualizacionMarzo2021.pdf

Thank you for mentioning this report.

L144-146: According to the AEMET´s historical cold spells technical note cited above, during the winter of 1999 there were no cold spells.

Thank you for mentioning this report.

L149: As mentioned in AEMET´s technical note on Filomena, at the end of December and the beginning of January there was a cold air advection from the north over the Iberian Peninsula, but just before the snowstorm, the cold air mass over the Iberian Peninsula probably was stagnated, with an intense nocturnal radiative cooling according to clear skies, calm or light winds and long nights. So, I think that the retrotrajectories study is not very interesting.

Since we will compare this event to other events, we decide later whether to include the trajectory analysis or not.

L184: and perhaps the downstream trough?

We do not think that the WCB directly influences the downstream trough. Potentially, it can indirectly influence the downstream trough by wave breaking.

L186: perhaps anticyclonic Rossby wave breaking (RWB) process?

This is what we mean.

L187: According to AEMET´s technical note, Filomena was nominated on 06 January. Heavy precipitation and strong winds affected the Canary Islands.

The exact timestep of cyclogenesis is not easy to determine, because Filomena resulted from a decaying low.

L193-194: According to AEMET´s technical note, in the medium and high levels analysis chart at 00 UTC, there is only one jet, the polar jet associated to the trough, and the Mediterranean convection is located on the left exit region of the jet.

In Fig. S1, two jets exist near the Iberian Peninsula. One is located to the northeast of Spain, which is the polar jet. We think that the jet south of Spain is the subtropical jet, because at this location, no PV gradient at 320 K is visible, which is a prerequisite for the polar jet.

L203: The trough evolves to a cut-off low, with a closed cyclonic circulation on 09 January.

We will check this with our dataset.

L205-206: I think it can be seen more appropiately with the soundings of Murcia and Madrid, or with a high-resolution model, instead the 0,5º ECMWF ERA-5 reanalysis.

We will not use sounding data, because they are only point observations. And for consistency, we stick to ERA5 reanalysis.

L217-219:  As mentioned before, the Iberian Peninsula has a complex orography, so the weather patterns related to heavy snowfall are different in every region:

Climatological characteristics and synoptic patterns of snowfall episodes in the central Spanish Mediterranean area.
(Nuñez et al, 2016)
https://rmets.onlinelibrary.wiley.com/doi/10.1002/joc.4645

Characterisation of snowfall events in the northern Iberian Peninsula and the synoptic classification of heavy episodes (1988-2018). (De Pablo et al., 2020)
https://rmets.onlinelibrary.wiley.com/doi/full/10.1002/joc.6646

The two referred episodes are quite different to Filomena from a synoptic point of view. The former one corresponds to a stationary low-pressure area over the Iberian Peninsula and the latter a deep polar low with a quick southward movement.
The weather patterns associated with snowfall in the meridional plateau are well known, a similar amount of snowfall (not in exactly the same region than Filomena) occurred on 4 of December of 1997.

Many thanks for the citations. We will include them when comparing snowstorm Filomena with other events.

L220: What do you consider lowlands? To compare different snowfall episodes it must be set the snow level. What was the snow level in these episodes?

Lowlands was indeed a bit unclear. We will reformulate this sentence.

Figure 2: Please, cite the source, ERA-5, it must be clear that this data are not real observations.

We already cited the data in Sect. 2.1, however, we can repeat this in Fig. 2.

Figure 6: In order to clarify the graphic, perhaps the isolines of 1 hour accumulated precipitation and snowfall can be deleted, furthermore, I think that information it isn´t relevant.

We will adapt the Figure.

Figure 7: 12-h or 24-h total accumulated precipitation seems more appropiate than 1-h total accumulated precipitation. The potential temperature anomaly with respect the model climatology is not relevant in this case, with a well-defined frontal zone.

We will adapt the Figure.

Figure 8: It could be better to choose another cross section, according to the air masses movement, from southwest to northeast.

We will better motivate why we show the cross section along 3°W.

**Reviewer 3**
In this paper an attempt is made to investigate the large-scale dynamics of a cold spell in Spain along with a following snowstorm and to examine their characteristics on a climatological basis. The paper deals with an interesting topic and the authors have investigated it to some extent. However, I have the following queries:

The study provided an analysis of basic large mechanisms that are well known for the occurrence of snowfall in the Mediterranean region. From this point, the study does not contribute any new knowledge. The only interesting feature I found is the analysis of the role of WCB. I think that the study should include mesoscale processes that facilitated the snowfall. Alternatively, the authors should perform a full analysis of other similar studies and comparison among them to exhibit similarities and/or differences. For instance, the authors refer briefly two cases in section 6 in comparison to Filomena, with obvious orographic forcing without further discussion. Furthermore, a comparison of the snowfall with forecast and investigation of possible forecast failure would be an interesting topic.

We will follow the second suggestion of the reviewer and will provide a full analysis of other events, e.g., long-lasting cold spells, heavy snowfalls, or the combination of both. However, we will not analyse the predictability of this event. According to the second reviewer, the forecasts issued by AEMET captured the event quite well.

The authors employed short term ECMWF forecasts for precipitation and snowfall during the examined period. Furthermore, they employed ERA5 reanalysis data with resolution 0.5 for the climatological part of their study during the period 1979-2021. I suspect that they employed the ERA 5 reanalysis data for plotting θe and PV (Figures 6, 7 and 8). However, they performed comparison of the specific snowfall event and the cold spell with other events employing different datasets, probably providing biases in their results (e.g section 2.3 or 4). I am wondering why they did not use ERA5 reanalysis data throughout the whole study.

We think that there is a misunderstanding. We use ERA5 reanalysis for the whole study. Accumulated variables, like precipitation, always come from short-range forecasts in the ERA5 dataset (Hersbach et al., 2020).

The analysis of trajectories was useful only for the part of WCBs. The source of the cold air mass can be easily seen from the synoptic analysis. More details should be provided about the use of the LAGRANTO model on a climatological basis.

As suggested by the reviewer, we will perform a full analysis (including trajectories) for other events. We will then decide whether we leave or remove the trajectory section. We will better explain, how we use LAGRANTO for the climatological analysis.

There are some points where there is no justification. For instance: the anticyclonic wave breaking (Fig 6e), the content of the clouds (Figure 8).

We will add more details on the anticyclonic wave breaking and will discuss the cloud water variables more thoroughly.

Observational data (station data, radiosonde data or radar data) should be incorporated to justify findings from the reanalysis plots.

We will not incorporate observational data since this is beyond the scope of our study. Our aim was to analyse the synoptic-dynamic processes leading to this event and compare this with other events and climatology. For that, we need spatially, temporally, and physically consistent data, which in our opinion is only fulfilled with reanalysis data.

More specific comments

Figures 6-7: These figures contain many details (coloured parameters, contours, labels) and it is difficult for the reader to follow
Figure 6: the letters L1, L2, F should be annotated what they stand for

We will adapt both figures as suggested.